# OpenReview forum: "GuardVal: Dynamic Large Language Model Jailbreak Evaluation for Comprehensive Safety Testing"
_ICLR.cc/2025/Conference — ICLR 2025 Conference Withdrawn Submission_

### Official Review · Reviewer_aJ5F · 2024-11-01

**Soundness:** 2
**Presentation:** 1
**Contribution:** 2
**Rating:** 3
**Confidence:** 4

**Summary:**

The paper presents a novel technique for evaluating dynamic jailbreaks in LLMs, using a role-based approach with four roles and four phases.

Phases:
	1.	Prompt Creation: The Translator creates prompts, which the Generator then refines for added complexity.
	2.	Jailbreak Testing: The prompts are tested on the model.
	3.	Evaluation: The Evaluator assesses the model’s responses to these prompts.
	4.	Refinement: The Generator adjusts prompts based on feedback, with the Optimizer overseeing to prevent stagnation.

A final model-relative ranking is produced based on a score combining both the model’s defense and attack responses. Results are benchmarked against existing rankings.

**Strengths:**

- The method is clearly presented but not very original, as other existing works have already implemented dynamic safety and jailbreak evaluation.
- The evaluation is robust and well-conducted, and the results presentation is clear and comprehensive.
- The prompt generation aligns with international standards, enhancing the method’s reliability and applicability across diverse contexts.

**Weaknesses:**

- The introductory and background sections are overly detailed, occupying substantial space before introducing the method almost on page 5. I'd resume these sections to around 2 pages to allow for a more detailed explanation of the core methodology.
- The Optimizer component, crucial in the evaluation pipeline, lacks sufficient explanation. I would better clarify its role and importance within the process.
- While the reason to combine attacking and defensive scores is explained, I don't understand this approach's rationale. Additionally, as the attacking score includes generation and evaluation, differentiation of specific model capabilities becomes complex. Although presenting both attacking and defensive metrics is important, I'd keep them separated.

**Questions:**

1. Could you further explain why the attacking and defensive scores are combined into a single metric? Specifically, how does this approach contribute to a comprehensive assessment of model capabilities?

2. Could you clarify how each attacker's role contributes to the score, or alternatively, consider separating them for clearer interpretation?

3. Could you explain more about how the evaluator decides whether a jailbreak is successful? Have humans overseen this process? What has been found?

**Details Of Ethics Concerns:**

The paper has the potential for misuse and production of harmful content. I'd move the ethical statements before in the paper.

---

### Official Review · Reviewer_bHvX · 2024-11-02

**Soundness:** 3
**Presentation:** 3
**Contribution:** 2
**Rating:** 5
**Confidence:** 4

**Summary:**

Current jailbreak datasets and benchmarks struggle to fully address the challenges of the evolving nature of LLMs. This paper introduces GuardVal, a new protocol that dynamically generates and refines jailbreak prompts based on the defender LLM's state. The authors propose a new optimization method that mitigates stagnation during the jailbreak prompt refinement process. Based on this method, this paper conduct a systematic study on jailbreak evaluation, opening future research directions.

**Strengths:**

1. The motivation is great and clear: The fixed nature of existing jailbreak datasets and benchmarks results in insufficient evaluation of consistently updated LLMs.

2. This paper is easy to follow.

3. Although not perfect, the metric Overall Safety Value is interesting and inspiring.

**Weaknesses:**

1. The rationale behind the overall safety value, especially the offensive capability. Although highlighted in Discussion, this reviewer is concerned about the rationale behind the design of offensive capability. Specifically, according to Equation 1, we can conclude that if an LLM's attack capability is stronger, then its OSV value will be worse. The authors should articulate a deeper rationale for such a design, e.g., why an LLM's ability to attack a target model can be correlated with its own security, and are they negatively or positively correlated?

2. The technical contribution of this paper is weak. As a benchmark paper, technical contribution is not a core evaluation indicator. However, the attack framework (like the design of role-play paradigm, including Translator, Generator, and Evaluator) has been documented in existing papers, like PAIR and TAP, and other jailbreak papers based on genetic algorithms [1, 2]. This makes this paper more like a comprehensive experiment based on these studies. The authors should clearly demonstrate the difference between this paper and existing works in terms of the attack/evaluation methods.

3. Lack of sufficient analysis of experimental results. Some results in Table 1 are counterfactual. For example, as an attacker, GPT-3.5 performs well in all LLMs, except the Vicuna. However, Vicuna is not a well-defensed LLM. Such data in the same row seems very unintuitive and the authors should provide more analysis. The same situation occurs with the use of LLaMa2 to attack Gemini.

[1] GPTFUZZER: Red Teaming Large Language Models  with Auto-Generated Jailbreak Prompts.
[2] OPEN SESAME! UNIVERSAL BLACK BOX JAILBREAKING OF LARGE LANGUAGE MODELS.

**Questions:**

1. See weakness 1: The rationale of the OSV metric.

2. See weakness 2: The technical contribution of this paper or the difference from this paper between existing works.

3. See weakness 3: Lack of sufficient analysis of unintuitive results.

**Details Of Ethics Concerns:**

None.

---

### Official Review · Reviewer_rYQk · 2024-11-03

**Soundness:** 3
**Presentation:** 3
**Contribution:** 3
**Rating:** 8
**Confidence:** 3

**Summary:**

The paper proposed GuardVal, a dynamic LLM jailbreak evaluation method that generates and refines jailbreak prompts iteratively to get credible and representative evaluation results. It employs a role-playing mechanism in which LLMs attempt to jailbreak other LLMs, ensuring the evaluation evolves in real time and adapts to the capabilities of the defender. The authors leverage proposed evaluation protocol to conduct the systematic study on jailbreak evaluation of SOTA LLMs.

**Strengths:**

1. Originality: the paper proposed a novel protocol to dynamically evaluate a group of LLMs in their capacity to defend jailbreak attacks by asking LLMs to generate jailbreak prompts for each other.
2. Quality: The method seems to execute well in experiment and have good result that matches existing LLM jailbreak benchmark.
3. Clarity: The paper is well-written and key information like prompt is provided in appendix.
4. Significance: The paper addressed the limitation of traditional human-labor-based methods and task-specific automated generation methods on jailbreak benchmark, contributing to the field in the long term.

**Weaknesses:**

1. The calculation of Overall Safety Value seems to be a bit arbitrary. Is Offensive Capability considered as important as Defensive Capability in the formula? Will LLMs with good offensive capability get advantages that are more than expected? Is there any correlation or consistency between defensive and offensive capability of LLMs? Analyzing values and ranking with either one capability respectively may help empirically "justify the relationship between offensive and safety".

**Questions:**

1. Asking LLM to generate jailbreak instruction for another LLM is offensive and dangerous itself, though I can see that the prompt template is cleverly designed so the intention is not obvious. But I still wonder if any LLM refused to respond as Translator/Generator due to its own safety limitation and how it may affect the results?

---

### Official Review · Reviewer_Kq8G · 2024-11-04

**Soundness:** 2
**Presentation:** 2
**Contribution:** 2
**Rating:** 3
**Confidence:** 5

**Summary:**

GuardVal presents three contributions: (i) a taxonomy of desiderata for jailbreak evaluation, focusing on a need for benchmarks to be able to evolve along with the models being jailbroken (ii) a jailbreak generation approach based on prompt refinement, with an additional optimizer that monitors for stagnation and tries to ensure diversity in generated attacks. (iii) a safety analysis of contemporary LLMs across 10 safety domains using the proposed methodology. The jailbreak generation approach relies on LLMs interacting with each other while adhering to various roles such as Translator, Generator, Evaluator, and Optimizer. All of these roles are evoked through few-shot prompting. Key observations from the evaluation of LLMs include (i) stronger jailbreaks are generated when more rounds of refinement are done in the generation process (ii) GuardVal results often disagree with existing static benchmarks (iii) some LLMs are better than others in the Attacker/Optimizer roles for attacking other LLMs.

**Strengths:**

1. The paper makes some useful contributions for subsequent work, such as a list of desiderata for jailbreak evaluation, and a dynamic jailbreak prompt generation approach, with a somewhat novel optimizer module to ensure no stagnation in generated attacks.
2. The optimizer design is also somewhat innovative, drawing inspiration from the Adam optimizer to maintain estimates of the change in responses over time, and converting these estimates to different sets of natural language prompts provided to the optimizer.
3. The authors have succeeded in making their OSV metric mostly agnostic to test set difficulty, since LLMs are evaluated on how easily they are broken by other LLMs and how easily other LLMs can break them.
4. The benchmarking performed is extensive, testing a large number of models on 10 different safety domains. I also found the discussion insightful, with thought being put into the different ways GuardVal can be used in practice.

**Weaknesses:**

1. In my opinion, presentation is one of the biggest issues holding this paper back. The background and OSV sections feel too verbose, and the jailbreak generation method is barely described in the main section of the paper.
2. There are other methodological concerns as well. The paper claims that the Optimizer results in a more diverse set of jailbreak attacks, but no analysis is presented confirming this hypothesis.
3. The jailbreak generation method is not compared to other methods in recent literature, such as GCG (https://arxiv.org/abs/2307.15043) and PAIR (https://jailbreaking-llms.github.io/).
4. The paper mentions that an uncontaminated test set is crucial for safety evaluation. However, the proposed method does not guarantee this: LLMs can simply verbatim memorize jailbreaking scenarios in their training data and regurgitate them during synthesis. This is especially true for recent LLMs, as large subreddits are now dedicated to jailbreaking LLMs (r/ChatGPTJailbreak for example).
5. There are also design choices in the jailbreak generation pipeline that require more explanation. Why is the attacker LLM allowed to decide what the rejected response should be? This is standardized in most recent literature. Why is semantic similarity used in scoring jailbreaks? Harm evaluation is a complicated problem, and most automatic evaluation approaches use sophisticated prompting to determine if generated responses contain harmful content. There is also no analysis of the alignment of the Evaluator responses with human preferences, making the results less trustworthy.
6. The number of rounds required for an attacker LLM to jailbreak the defender LLM can vary substantially between test cases. This variance is not accounted for in OSV. Whenever a new LLM is introduced, OSV requires using this LLM as an attacker and defender against every other LLM in the benchmark, limiting its practicality, while also recomputing OSV numbers for every other LLM in the benchmark.
7. Some more motivation is also needed for why the capability of an LLM to jailbreak other LLMs is a factor in deciding how safe it is.

**Questions:**

1. “Developing model-specific methods would enable increasingly complex and tailored evaluations, ensuring that test data evolves alongside the LLM’s capabilities.”: this line requires some clarification.
2. “Task-Specific Focus and Lack of Model-Specific Evaluation:” this part of the background requires citations from existing literature to validate that this is a problem. There is also a substantial space of model-specific jailbreak generation methods in existing literature, such as GCG, PAIR and h4rm3l (https://arxiv.org/abs/2408.04811).
3. “Studies have shown that simplistic prompts with bizarre sequences are easily detected and fail to expose the true weaknesses of LLMs.”: this requires a citation.
4. There is a repeated paragraph in Page 6 ("Potential Concerns on Handling Outliers")
5. What are the "fragments" referenced in the generator description?
6. More details are required regarding how the change in response between iterations is computed for the optimizer.

---

### Note · Authors · 2024-12-04

I have read and agree with the venue's withdrawal policy on behalf of myself and my co-authors.